# The Assessment of Greyfields in Relation to Urban Resilience within the Context of Transect Theory: Exemplar of Kyrenia–Arapkoy

Vedia Akansu [1,*] and Aykut Karaman [2]

1   Faculty of Architecture, Near East University, 99138 Mersin, Turkey
2   Department of Architecture, Faculty of Engineering and Architecture, Altinbas University,
    34218 Istanbul, Turkey
*   Correspondence: vedia.akansu@neu.edu.tr

**Abstract:** Greyfields are construction sites that emerge from the expansion of cities towards rural settings. They are unused structures in settlement areas that negatively impact the habitats and lead to ecologically, economically, and socially problematic zones. This study aims to examine the Greyfield problem, which emerges as one of the outcomes of urban sprawl, within the context of Transect Theory and urban resilience. We analyze the Greyfield problem in the Arapkoy rural settlement, which is located along the north coastline of Kyrenia, Cyprus. This study presents the impact of Greyfield sites on environmental, social, and economic values within the framework of Transect Theory. Thus, a road map for the redevelopment of Greyfields into public use is put forward to be used for future planning activities, which is a necessity in enabling urban resilience.

**Keywords:** Greyfield; urban transect; urban resilience; rural areas; ecology

## 1. Introduction

Resilience is the capacity of a system to maintain its original organizational structure in terms of unity and identity over time. It is about absorbing external shocks due to its self-organizational capacity. In other words, the concept of resilience refers to the capacity of a city to recover from a wide range of shocks and risks [1–3]. Due to its wide potential applicability in local practice, urban resilience provides a popular paradigm for urban planning and policy making at different spatial scales, including cities, regions, and neighborhoods [4,5]. Some common elements that can be identified in ecological, economic, and social resilience are the notion of memory, conservation, stability, correlation, and feedback [6]. A resilient city is more talented in adapting to an increasingly complex economic environment. It will successfully cope with sudden natural disasters and social emergencies, better protecting the health of the eco-system, and meeting information requirements [7–10]. With the acceleration of urban agglomeration, urbanization has become a complex system with multiple, closely linked factors that impact themselves and the outside world [11–15]. The concept of resilience is a holistic planning approach in the sense that it addresses the areas to be protected together with the dynamics associated with development areas. It assesses ecological, economic, and social data together and intervenes from urban to rural areas with dynamic tools [16]. The effect of irregularity and destabilization in urban and rural areas increases the probability of problems. Irregular structures, particularly those built in geologically unsuitable areas, and inadequacy of planning are fundamental factors in the occurrence of issues [17]. Transect Theory aims to organize all of the elements in the environment and includes urban and rural settings and centers. It is also a system that puts forwards different settlement types and concepts in varied urban densities [18]. Therefore, besides transportation axes, areas with a high density of buildings and their uses, design features of facades, masses, public spaces, junctions, car

parks, pavements, street silhouettes, lighting elements, green areas, and landscape elements are identified.

Furthermore, attention has been paid to ensuring that all urban elements in the system are in the appropriate place and continuous [19,20]. In this regard, the urban–rural scales classification scheme offers a settlement proposal for city centers and natural open spaces in rural areas. Thus, suburbanization is crucial in improving cities that insensibly expand with decentralization.

Greyfield sites are found in built-up urban areas and are one of the best choices for redevelopment because they utilize sites that are no longer profitable, are not significantly contaminated, and reduce financial risks. They are usually found in the form of old retail malls, office parks, abandoned municipal properties, and parking lots [21]. Small-scale, piecemeal, fragmented, suboptimal small-lot subdivision is spreading like a virus through Greyfield suburbs with high redevelopment potential, removing up to 50% of private green space and blocking prospects for better designed, regenerative, precinct-scale, medium-density "missing middle" redevelopment [22,23]. Greyfields typically emerge as a result of urban expansion. These are fields that are left unused, which are architecturally old-fashioned, have insufficient or unusual infrastructure, lack redevelopment capital, and create a negative impression [24]. In other words, "Greyfield" is a term used to describe the extensive band of aging, occupied, residential tracts of inner and middle suburbs that are physically, technologically, and environmentally obsolescent, and which represent economically outdated, failing, or under-capitalized real-estate assets [25]. These problematic areas cause ecological, economic, and social problems that are generally observed in suburban settlements [26]. They emerge as problem areas that pose a threat to the resilience characteristics needed for livable cities. Although they are unused areas, Greyfields cause the loss of rural features (i.e., agriculture and animal husbandry) in lands that are opened for settlement, deterioration of topography, and destruction of ecology. They also change the morphological structure of urban and rural areas.

Abundant literature exists highlighting the benefits derived from Greyfields in the USA and Australia as residential and commercial exemplars [25–41]. However, this literature has not dealt with the externality of Greyfield redevelopment, and there are a lack of case studies about the redevelopment of Greyfields in European and Asian countries [42]. In the Australian and American contexts, Greyfields are described as retail malls or commercial buildings that failed to succeed and which are usually located in suburban areas characterized as having potential for redevelopment. Although heavily described as "derelict buildings" in the literature, Raamos (2011) broadly defines Greyfields as unproductive urban spaces that can yield economic, social, and environmental benefits [43]. Within the context of North Cyprus, although the structures that are discussed in this paper are mainly derelict buildings, the majority of them were constructed in a large lot with the purpose of built-to-sell, thus can be considered commercial buildings. Therefore, for this study the "Greyfields" term has been adopted. These buildings are left uncompleted at different stages of construction, or they are abandoned due to the political or financial difficulties that construction companies experience. Thus, this unique characteristic of Greyfields in North Cyprus contributes to the relevant literature.

Another theme that is important for the study is the Transect Theory. This theory is a cut or path through part of the environment, showing a range of different habitats. Biologists and ecologists use transects to study the many symbiotic elements that contribute to habitats, where certain plants and animals thrive [44]. The concept of the transect in the urban environment is embedded in traditional cities, as a tool used informally by humanity [45]. Andres Duany and other urbanists applied this concept formally to human settlements and used it in planning beginning in the late 1980s [44,46]. The idea has evolved into an organizing theory, permeating since about 2000 with the consideration of new urbanism [44,47]. In this regard, Transect Theory has been adopted as the theoretical framework for study as it is a system that engulfs all of the elements of urban, rural, and suburban areas and it supports the planning approaches that protect ecological, economic,

and social values. Therefore, these characteristics make Transect Theory a more useful tool when it is used along with urban resilience. Within this context, this paper aims to examine the Greyfield problem, which emerges as one of the outcomes of urban sprawl, within the context of Transect Theory, and to analyze the Greyfield problem in the Arapkoy rural settlement in relation to urban resilience. The Arapkoy rural settlement is located along the north coastline of Kyrenia, Cyprus.

This article, following this brief introduction, is structured into the following sections: the methodology description, the presentation of the case studies, and the discussion of results, with some final remarks and future perspectives on the research. The main aim of this paper is to help contribute to the research on the Greyfield problem, which emerges as one of the outcomes of urban sprawl, within the context of Transect Theory and to analyze the Greyfield problem in relation to urban resilience, since the Greyfield is still a quite recent topic and in need of further development.

## 2. Materials and Methods

### 2.1. General Methodology

The framework of this study was developed based on the existing relevant literature, and data were collected through secondary acquisition of reports and a questionnaire. Due to the building dynamics and density, and existing Greyfield sites in both green and built areas, the Central and East Kyrenia settlements were chosen as the foci of this study. Secondary data such as reports provided numerical facts about the studied region, whereas questionnaires were applied both for residents and experts to minimize the disadvantage that comes from using secondary data. This way, the statistical data provided an understanding of the status in the regions examined, and questionnaire results depicted the current state and existing problems. The questionnaire was conducted over 6 months from the last quarter of 2016 until the first quarter of 2017. Ecological, economic, and social values of problematic Greyfield sites for urban resilience were evaluated within the scope of data collected by questionnaire with the stakeholders, including residents, non-governmental organizations, and relevant official regional institutions. In the conducted questionnaire, Arapkoy resident participants were referred to as "residents" and individuals such as architects, engineers, and city planners with active duties in public organizations and non-governmental bodies that are related to the region were referred to as 'experts'. Questionnaires were completed by 67 individuals consisting of two groups: one group including 31 experts, and the other group including 36 residents. The questionnaire was conducted over 6 months. It started during the last quarter of 2016 and was completed during the first quarter of 2017. The questionnaire was designed in two main sections: the first section was designed to put forward the five most important problems in light of the residents' and expert participants' views, and the second section was designed to put forward the ecological, economic, and social status describing city resilience in percentages. In the first section, 20 factors were determined. These factors included: transportation, roads, security, increasing crime rates, street lighting, car parks, water mains, sewer system, air pollution, garbage and solid waste, unused and unnecessary stock of housing, unplanned construction, fields for activity, health services, unemployment and lack of employment, lack of infrastructure services, a rapid increase in population, lack of legal regulations, visual pollution, noise, and limited green fields in urban areas. The second section consisted of 10 questions with a 5-point Likert scale, where participants were asked to rate their agreement levels with the statements. Statements were as follows: Greyfield sites contribute to the future of the regional economy; ensure protection and transfer of the region to future generations; contribute to the development of tourism, create pressure on the natural environment; prevent protection of environmental and ecological values; the existence of housing above the carrying capacity of the area leads to environmental pollution; the area is not guided with planning; Greyfield sites diversify the locals' income sources; Greyfield sites limit the agricultural activities, which are a primary source of income; they create regional risk areas in terms of urban resilience (i.e., natural disasters such as flood and

earthquake, technological risks; human originated risks and obstacles to the development of urban and rural areas.

The residents were chosen randomly among Arapkoy residential users. On the other hand, the experts were chosen randomly among those who work in Arapkoy and know the region. As Arapkoy has a population of 542 according to the 2011 census [48]. it was suggested to be a sufficient sampling size for the research. For example, a study aiming at measuring the environmental attitudes among local people via a user survey in Guzelyurt, North Cyprus (having a population of 18,946) used a random sample of 60 residents [49]. In addition, an article cited in our study involving a survey with a sample of 367 focused on Helsinki with a population of almost 642,000 residents.

Data on dwelling count and census results were derived from the State Planning Organization. Because the last official census was in 2011 and the results were comparatively old, statistical data from the Cyprus Turkish Building Contractors Association (CTBCA) were used, covering the 2011–2016 period. The authors acknowledge the relatively aged nature of the secondary data and the lack of other comprehensive reports.

Based on the sketch studies made for the relevant zones, a suitable transect schema (Figure 1) was created by presenting the building density, usage status, and purpose of construction.

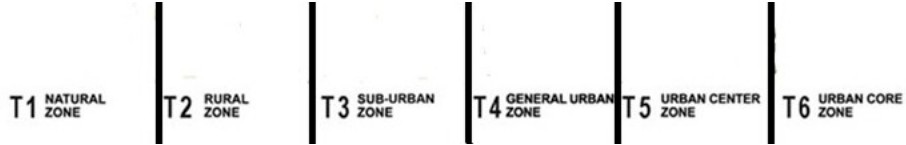

**Figure 1.** Transect schema structure.

The transect approach is an analytical method and a planning strategy. It can be formally described as a system that seeks to organize the elements of urbanism (street, land use building, lot, and all of the other physical elements of the human habitat) in ways that conserve the integrity of the different types of urban/rural environments. These environments can be viewed as variations along a continuum that range from rural to urban. Along this continuum, human environments vary in their level of urban intensity. Cleaving to this system of organization, urban open spaces are saved in their urban state, while rural settlements are preserved in their rural state, and the mixing of elements—a rural element in an urban open space and vice versa—is avoided [50].

The transect schema is divided into six zones: natural (T1), rural (T2), suburban (T3), general urban (T4), center (T5), and urban core zone (T6) [19,51]. According to Emily Talen, each transect zone is an immersive environment, a place where all of the elements reinforce each other to produce and consolidate the character of a specific place [50].

The significance of this approach comes with its part as a taxonomic machine. It is like a new zoning system, compared to the conventional separated-use zoning systems. The new system provides the base for real neighborhood structure, which requires walkable streets, mixed-use, transportation options, and dwelling diversity. The T- zones vary by the rate and position of the intensity of their natural, built, and social factors, in addition to their part in the creation of an environment that could be perceived as immersive [52]. Also, the creation of a more environmentally conscious sprawl, the preservation of natural and agricultural land, the creation of flexible communities, balancing every aspect of the city, enabling diversity, and the management of urban development raise the quality of the erected environment in general and produce better places to live [47,51].

Following this, a comparative analysis was used to assess the urban resilience of the suburban and rural regions where the heavy impact of construction was observed. Arguments have also been presented on how the comparative analysis of urban resilience can add value to future planning activities.

*2.2. Study Area*

Cyprus (Lat 34.33 and 35.41 N; Long 32.23 and 34.55 E) has a typical Mediterranean topographical and architectural building style. The topographic structure is influential in the formation of ecological and social regions. This topographical impact indicates that individuals belong to social groups, and they live in particular urban or rural residential areas, which are insulated according to their environment [53]. The transportation axes are also shaped as a result of the topographical structure [54].

However, Cyprus Island has been divided into north and south since 1974 due to ethnopolitical reasons. The effects of construction density have not been evident until the early 2000s in North Cyprus, the context of this study, and it is observed that the social, cultural, and ecological values have been preserved.

The "Annan Plan" prepared by the United Nations (UN) in 2001 was discussed to provide a comprehensive solution to the Cyprus conflict. Because it was proposed as a resolution plan by the UN, it found credibility in the international arena. Thus in 2007, this plan put North Cyprus into a "rapid structuring" process, especially with the assurance of eliminating some of the uncertainties about property issues [55]. Parallel with these developments and changes, progress was observed in construction, particularly in the Kyrenia region in 2003–2004.

A demand mainly from the UK, North Europe, and other countries was observed in the northern part of Cyprus, especially in the Kyrenia region [56]. Thus, the construction sector and urban architectural environments started to develop and spread toward the rural areas. Intensive housing was observed primarily on the northern coastline of Kyrenia. The different building styles such as villas, apartments, and duplex houses alongside hotel buildings, which showed rapid increase, brought diversity to the architectural environment in Kyrenia and other regions [57]. The increasing demand in the construction sector positively impacted the economy, and people started to leave their existing jobs and enter the construction business. Houses with a similar style were built with the aim of economic gain, without considering customer demands and needs. However, legal regulations related to the construction sector "Streets and Buildings Chapter 96 of the Laws" were insufficient. Construction permits were given to every plot of land with road access, including pasture, agricultural land, and qualified land [55]. However, the construction sector collapsed due to the surplus supply of houses, and the construction companies experienced economic problems alongside title deed problems, which were rooted in the political conflict on the island.

Today, some of these buildings are left half-completed, and some of them are completed but unsold. They create ecologically, economically, and socially problematic areas and cause problems in ensuring urban–rural resilience.

## 3. CTBCA Report

To resolve the problems experienced, the CTBCA assessed the situation with statistical studies aimed at determining the new housing densities of the northern Cyprus settlements based on KADEM data, which is a private research company. The household data of the census results of 2007 and 2011 were examined. In the CTBCA report, Cyprus's northern coastline was investigated in four different regions.

- Kyrenia–Eastern Region (Kyrenia Karaoglanoglu in the East, Sadrazamkoy in the West, Besparmak Mountains in the South)
- Kyrenia–West Region (Kyrenia Karakum in the West, Buyukkonuk Junction in the east of Kaplica)
- Famagusta–Bogaz Region (Tuzla in the South, Bogaz in the North)
- Bogaz–Karpaz Region (Bogaztepe in the West, Dipkarpaz in the East)

Construction was investigated in terms of building categories such as houses, flats, and shops based on the regions. Statistical information on their development density is given below (Figure 2):

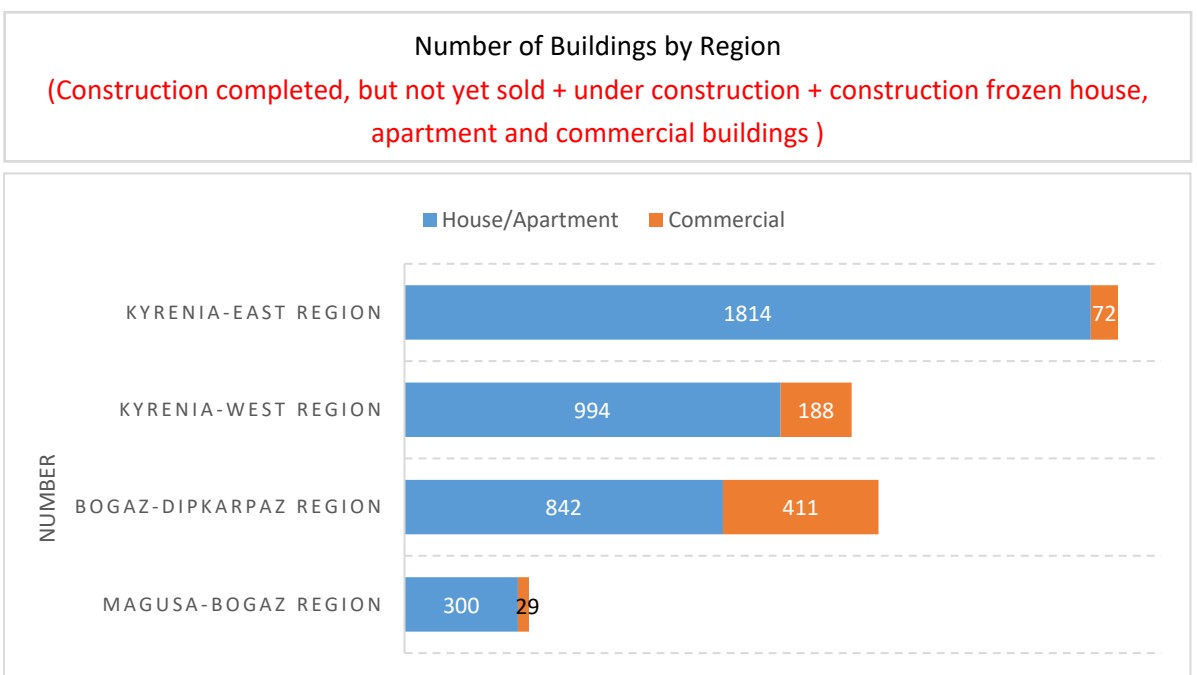

**Figure 2.** The number of buildings on the North Cyprus coastline [58].

Study on the construction count—phase 1 was compiled by CTBCA in 2011. The data indicated that the eastern coastline of Kyrenia was the most intense settlement with 1886 buildings; of those, 72 were commercial buildings. Furthermore, 86.9% of the 1886 buildings were built by construction companies. The significant majority of these buildings (90.9%) were constructed in the east settlements of Kyrenia (Figure 3).

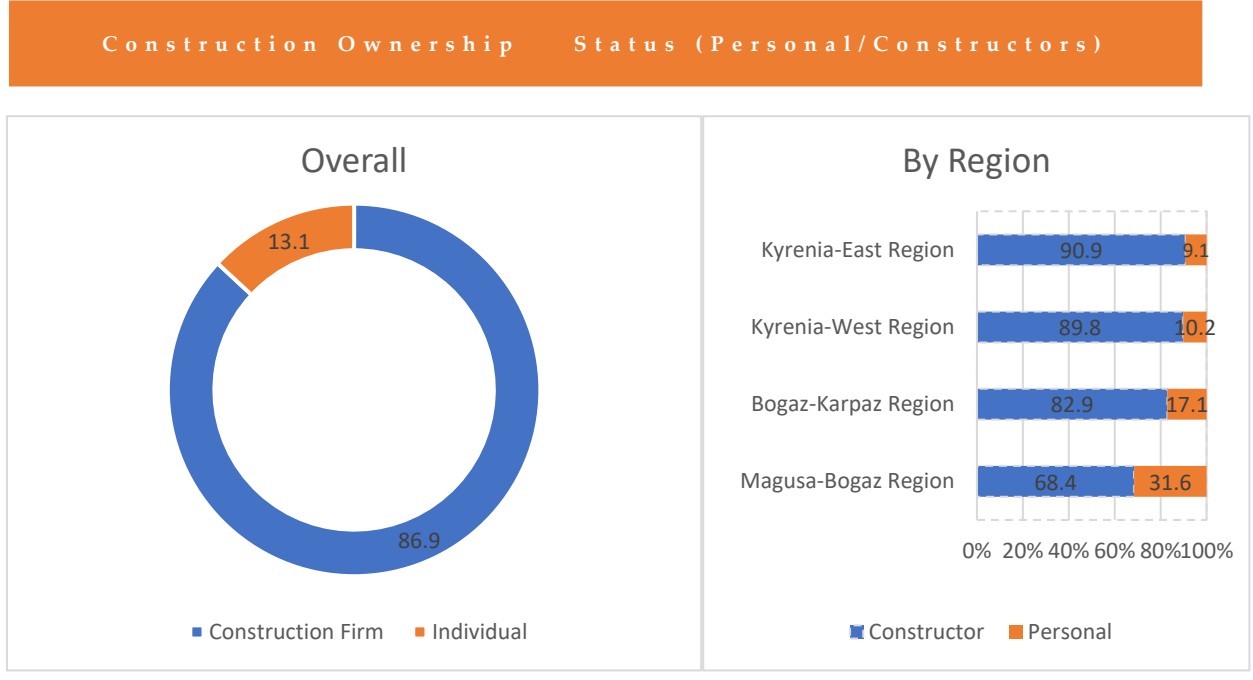

**Figure 3.** The assessment of the construction firms or individuals who built residences on the Cyprus coastline [58].

The inadequacy of the planning not only caused the irregular construction on the coastline, but it also revealed that there was an increase in the number of derelict houses.

Therefore, the location of the construction was assessed according to the regions. A total of 7.9% of the building constructions were halted, whereas 52.9% were still under construction, and 39.2% of building construction were completed but not yet sold (Table 1).

**Table 1.** Evaluation of the construction location according to the designated regions in North Cyprus [58].

| REGION | THE STATUS OF CONSTRUCTION BY REGION AS OF 2011 | | | | | | | |
|---|---|---|---|---|---|---|---|---|
| | Construction Completed | | Under Construction | | Construction Halted | | Total | |
| Magusa–Bogaz Region | 111 | 33.7% | 191 | 58.1% | 27 | 8.2% | 329 | 100% |
| Bogaz–Dipkarpaz Region | 418 | 33.4% | 710 | 56.7% | 125 | 9.9% | 1253 | 100% |
| Kyrenia–West Region | 486 | 41.1% | 649 | 54.9% | 47 | 4.0% | 1182 | 100% |
| Kyrenia–East Region | 809 | 42.9% | 910 | 48.2% | 167 | 8.9% | 1886 | 100% |
| Total | 1824 | 39.2% | 2460 | 52.9% | 366 | 7.9% | 4650 | 100% |

As mentioned above, while many residences were waiting to be completed in the construction phase, many of them did not have buyers and remain derelict. It was observed that the Kyrenia–East region, followed by the Bogaz–Dipkarpaz regions were affected by this situation the most. The longitudinal assessment presented in Table 2 covers the years between 2007 and 2011.

**Table 2.** Evaluation of the construction location according to the designated regions in North Cyprus [58].

| REGION | NUMBER OF CONSTRUCTIONS | | | | | | | |
|---|---|---|---|---|---|---|---|---|
| | (Construction completed, but not yet sold + under construction + construction frozen house, apartment, and commercial buildings) | | | | | | | |
| | 2007 | | | | 2011 | | | |
| | Under construction+ frozen | Construction completed, but not yet sold | | Total | | Under construction+ frozen | Construction completed, but not yet sold | | Total | |
| Bogaz–Dipkarpaz Region | 424 | 52.54% | 383 | 47.46% | 807 | 100% | 418 | 33.36% | 835 | 66.64% | 1253 | 100% |
| Kyrenia– East Region | 2812 | 81.81% | 626 | 18.21% | 3437 | 100% | 1077 | 57.10% | 809 | 42.90% | 1886 | 100% |

Table 2 presents statistical data on Greyfields in the Kyrenia–East Region settlement. There was a total of 3437 Greyfields in 2007 and 1886 in 2011. Today, new construction projects continue while the Greyfields remain idle.

Thus, grey-area formation typologies, which are discussed in regional development, are found dominantly in newly constructed parts of rural areas, which exist on city borders.

*Greyfield Statistics for Rural Areas*

The difficulties experienced in urban planning cause problems in rural areas as well. Planning studies that cover all urban and rural areas are needed for the preservation of rural characteristics and their transfer to future generations.

The new dwelling increase in settlements located east of the northern coastline of Kyrenia, where most housing is located, is as follows (Figure 4).

Figure 4 shows the number of new houses built between 2004 and 2007. New residences were built in Besparmak (59), Karakum (73), Alagadi (102), Karaagac (120), Esentepe (170), Tatlısı (202), Esentepe-Tatlısı (206), Arapkoy (300), Bahceli (548), Alagadi-Esentepe (874), Kucuk Erenkoy (983), and Catalkoy (1063). An increase in the number of dwellings among rural settlements between 2004 and 2007 was mostly observed in Arapkoy due to its proximity to the city of Kyrenia. Furthermore, it was determined that a total of 4700 new houses were built between 2004 and 2007 between the Karakum and Tatlisu areas in the East–West direction and between the north coast and the Besparmak mountain

in the North–South direction [59]. According to CTBCA data in 2007, the Kyrenia–East Region was consistent with statistical data from 2007, with 1263 (27%) of the houses built in the region having residents. Of the remaining 3437 (73%) houses, 2812 were under construction, and 626 of them were vacant although they were completed [59]. Figure 4 shows the status of vacant and occupied dwellings according to the regions. When vacant and occupied percentages were examined for 2011, it was evident that the majority of the constructions were uncompleted, and for those which were completed (1886), 73.1% were vacant (Figure 5).

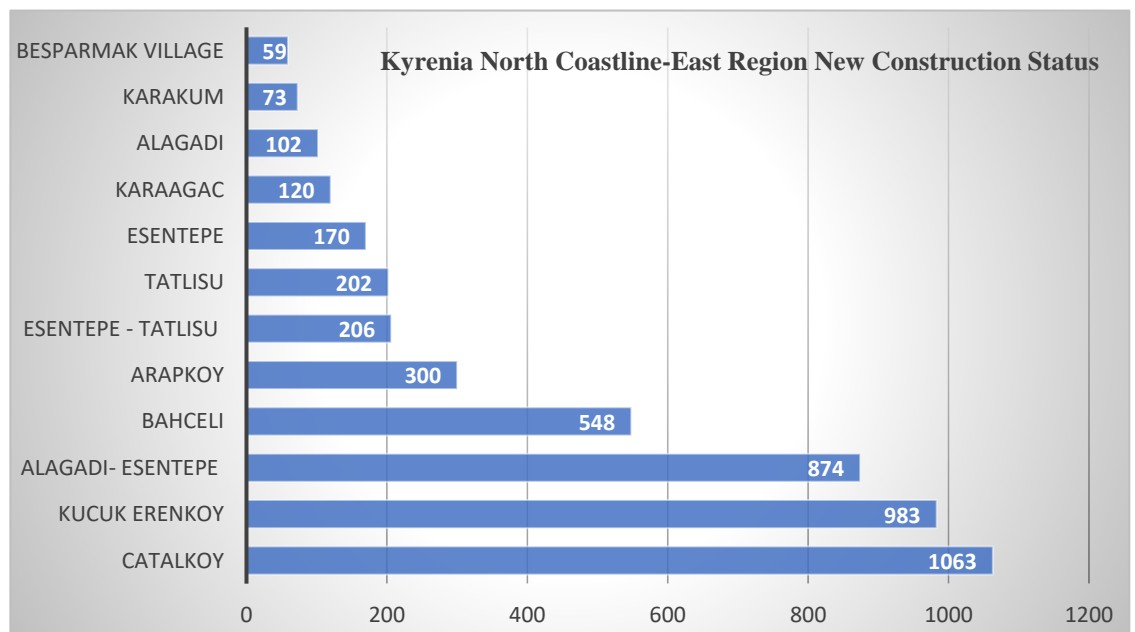

**Figure 4.** Kyrenia–East Region housing increase from 2004 to 2007.

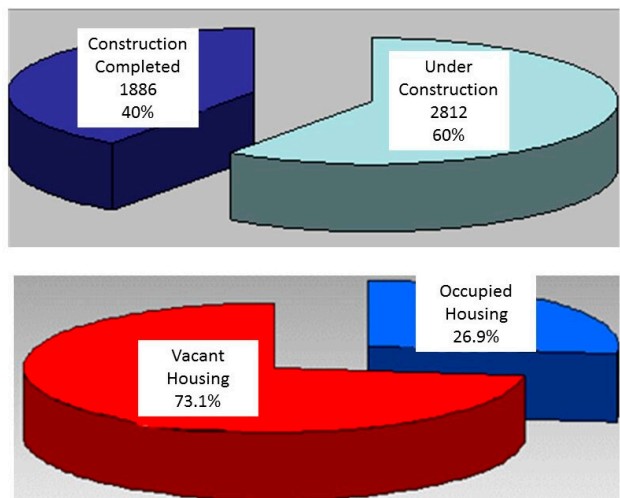

**Figure 5.** Vacant and occupied dwelling percentages in the Kyrenia–East Region as of 2007.

The inventory study of houses built after 2007 in the Kyrenia region was compiled according to the household count in the 2011 census results by the Cyprus Turkish Building Contractors Association [58].

The overall dwelling occupancy rate, which was 26.9%, reached the highest rate in Karakum with 60.35% and 50% in the region between Esentepe-Tatlisu. The percentage of vacant houses in Besparmak village was 100%, 91.8% in Kucuk Erenkoy, and 85.3%

in the Alagadi regions. However, a high number of abandoned houses were observed in Bahceli and Arapkoy in the east settlements of Kyrenia. The Arapkoy settlement was designated as a study area because it was close to Kyrenia city and was considered a grey area. Furthermore, it was determined that there were intensive residential construction and land subdivisions in the area. It was concluded that the investments in industrialization, tourism, and commercial or multi-purpose construction were at a very low level [58] (Figure 6).

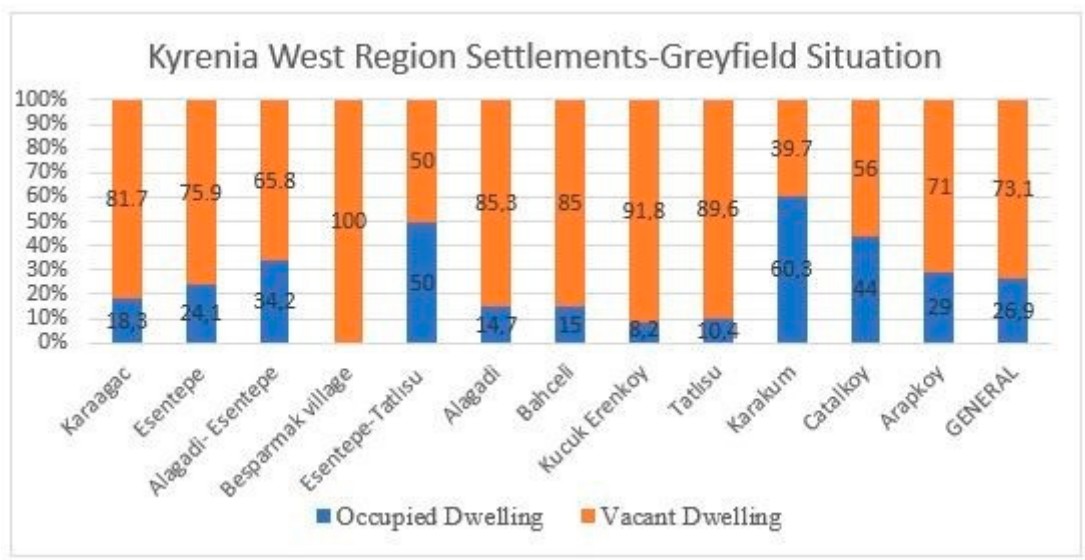

**Figure 6.** Percentages of occupied and vacant house settlements in the east of Kyrenia [58].

The inventory of houses built after 2004 in the Kyrenia region was compiled within the scope of the 2011 census results by the Cyprus Turkish Building Contractors Association [58].

The overall percentage of occupied dwellings was 26.9%. However, this percentage reached up to 60.35% in Karakum and 50% in Esentepe-Tatlisu. On the other hand, the rate of vacant dwellings was 100% in Besparmak village, 91.8% in Kucuk Erenkoy, and 85.3% in the Alagadi regions. It was observed that there were completed houses in the Bahceli and Arapkoy settlements (Figure 5), but they remained abandoned. In terms of the types of construction, it was determined that predominantly housing and plot parceling constructions had taken place in the region. It was concluded that the investments in terms of industrialization, tourism, and commercial and multi-purpose building were limited [60].

The construction information database booklet (2016) of the CTBCA stated that there was an increase in construction permit applications by 8% in Nicosia, 32% in Famagusta, 37% in Kyrenia, and 20% in the Guzelyurt district according to the 2015 data. On the other hand, there was a 38% decrease in the Iskele district. These figures suggested that the Kyrenia district experienced the highest increase in construction permit applications [60]. In the construction files recorded in 2015 across North Cyprus, the Kyrenia district ranked first with 716 files. Of the 716 files, 123 of the applications were completed, and 593 of them were ongoing as of 2016 [60].

## 4. Results and Discussion

### 4.1. The Rural Planning Issues and Resilience in Greyfield Sites

Urban resilience not only focuses on the resilience of large cities but additionally includes towns and smaller living areas [61]. Rural areas play an essential role in ensuring urban resilience. An evaluation index system of urban resilience was made using systematic, scientific, comprehensive, and availability principles, combined with the particular

development of the cities. This index system was composed of four sub-resilience systems: ecological, economic, social, and infrastructure resilience [62].

Unplanned construction and distorted urban settlement in Kyrenia led Greyfields to be frequently observed in rural areas. The Arapkoy settlement is crucial because it accommodated exemplars of the aforementioned problem.

Within the context of this research, resident and expert opinion was sought to determine the impact of disorderly construction on urban–suburban and rural areas on the east coast of Kyrenia. In this section, 20 factors were determined: transformation, roads, security, increasing crime rates, street lighting, car parks, water main, sewer system, air pollution, garbage and solid waste, unused and unnecessary stock of housing, unplanned construction, fields for activity, health services, unemployment and lack of employment, lack of infrastructure services, and rapid urbanization in urban areas. Of these, participants were asked to rank the five most important factors. The results are presented in Table 3.

**Table 3.** The five most important problems determined by resident and expert participants on the Kyrenia coastline.

| The Five Most Important Problems on the Kyrenia Coastline | | | |
|---|---|---|---|
| *RESIDENTS* | | *EXPERTS* | |
| 1 | Unplanned Construction | 1 | Unplanned Construction |
| 2 | Sewer System | 2 | Roads (Asphalt Paving and Pavements) |
| 3 | Air Pollution and Transportation (Insufficient Public Transport) | 3 | Transportation (Insufficient Public Transport) |
| 4 | Visual Pollution (Greyfield Sites) | 4 | Insufficient Infrastructure Services |
| 5 | Limited Green Fields in Urban Areas, Roads (Asphalt Paving and Pavements), Garbage and Solid Waste | 5 | Car Park |

Both residents and experts agreed that unplanned construction and insufficient public transport, asphalt paving, and pavements were the most important problems on the Kyrenia coastline.

The experts also identified inadequate infrastructure services and car parks, while the residents reported limited green spaces.

Further to this, a comparative analysis was provided to examine the opinions of both the residents and experts on the impact of Greyfield sites based on their responses to 10 questions on a 5-point Likert scale (strongly disagree to strongly agree). The questions determined during the survey were selected for the ecological, economic, and social maintenance of the Greyfield regions, including how Greyfield sites contribute to the future of the regional economy; ensure the protection and transfer of the region to future generations; contribute to the development of tourism; create pressure on the natural environment; prevent protection of environmental and ecological values; how the existence of housing above the carrying capacity of the area leads to environmental pollution; how the area is not guided by planning; how Greyfield sites diversify the locals' income sources; how Greyfield sites limit the agricultural activities, which are a primary source of income; how they create regional risk areas in terms of urban resilience (i.e., natural disasters such as flood and earthquake, technological risks; human originated risks and obstacle to the development of urban and rural areas). The results are presented in Table 4.

**Table 4.** Arapkoy Rural Area—Greyfield Site Assessments.

| Greyfield Sites-Related Questionnaire Questions | Arapkoy Rural Area Greyfield Sites Impact Assessment | | | | | | | | | |
|---|---|---|---|---|---|---|---|---|---|---|
| | Strongly Agree | | Agree | | Neither Agree Nor Disagree | | Disagree | | Strongly Disagree | |
| | Resident | Expert | Resident | Expert | Resident | Expert | Resident | Expert | Resident | Expert |
| Greyfield sites contribute to the future of the regional economy | 11% | 3% | 22% | 13% | 11% | 13% | 34% | 32% | 22% | 39% |
| Ensure protection and transfer of the region to future generations | 6% | 3% | 19% | 0% | 17% | 3% | 36% | 29% | 22% | 65% |
| Contribute to the development of tourism | 12% | 3% | 28% | 3% | 17% | 3% | 20% | 42% | 23% | 49% |
| Create pressure on the natural environment. Prevent protection of environmental and ecological values | 11% | 49% | 39% | 39% | 14% | 3% | 31% | 3% | 5% | 6% |
| The existence of housing above the area's carrying capacity leads to environmental pollution | 17% | 62% | 67% | 29% | 5% | 0% | 3% | 3% | 8% | 6% |
| The area is not managed with a plan | 25% | 65% | 47% | 29% | 19% | 0% | 3% | 3% | 6% | 3% |
| Greyfield sites diversify the locals' income sources | 8% | 10% | 31% | 10% | 28% | 26% | 22% | 22% | 11% | 32% |
| Greyfield sites limit agricultural activities, which is a primary source of income | 25% | 29% | 31% | 29% | 14% | 23% | 16% | 12% | 14% | 7% |
| They create regional risk areas in terms of urban resilience (i.e., natural disasters such as floods and earthquakes, and technological and human-originated risks). | 25% | 45% | 42% | 39% | 8% | 10% | 19% | 6% | 6% | 0% |
| Greyfield sites are obstacles to the development of urban and/or rural areas | 17% | 26% | 46% | 58% | 17% | 6% | 17% | 10% | 3% | 0% |

As it is stated in Table 4, there were items that the residents and experts did not agree with. Regarding economic issues, 56% of the residents and 71% of the experts disagreed or strongly disagreed that Greyfield sites "*contribute to the future of the regional economy*". A total of 94% of the experts and 58% of the residents stated their disagreement that Greyfields "*ensure protection and transfer of the region to future generations*". When participants were asked if they think Greyfield sites "*contribute to the development of tourism*", 28% of the residents agreed and 49% of the experts disagreed with that statement.

The questions that both experts and residents agreed on included "*existence of housing above the area's carrying capacity leads to environmental pollution*"(experts (62%) and residents (67%)). A total of 65% of the experts and 47% of the residents agreed that the Greyfield sites "*lack planned management*".

When the participants were asked whether "*Greyfield sites create regional risk areas in terms of urban resilience*", both experts (45%) and residents (42%) agreed with this statement, and 58% of the experts and 46% of the residents also agreed that "*Greyfield sites are obstacles to the development of urban and rural areas*". Also, regarding the statement "*Greyfield sites creating pressure on the natural environment and preventing the protection of environmental and ecological values*", 49% of the experts and 39% of the residents agreed with this statement.

The Arapkoy rural area Greyfield sites create pressure on the natural environment and prevent ecological values from being preserved. The participants also suggested that the Greyfield sites were not managed with a plan. This was partially due to the existence of housing above the area's carrying capacity, which also caused environmental pollution. As a result of the questionnaire, it was concluded that the Greyfield sites limited the residents' agricultural and farming activities, which was their source of income, created regional risk areas in terms of urban resilience, and caused obstacles to spatial development. These conclusions suggested that the Greyfield sites should be taken into consideration in planning activities that focus on enabling resilience, as Greyfield sites affect the environmental, economic, and sociological factors of a region. It would not be

possible to consider resilience and habitable settlements without taking into consideration the aforementioned problems while conducting planning activities.

### 4.2. The Urban–Rural Transect in Greyfield Sites

The transect scheme aims to organize the city with all of its details in the planning stage. It is also an effective method in bringing the Greyfields into use. By considering the grey areas together with the transect scheme, the assessment of the existing situation is introduced to transform grey areas into habitable areas. However, research has indicated that this problem arises as a result of urban expansion. As Transect Theory suggests, Greyfields are divided into regions and considered as a whole (Figure 7).

Urban–Rural Scale Classification Scheme (Transect Diagram)

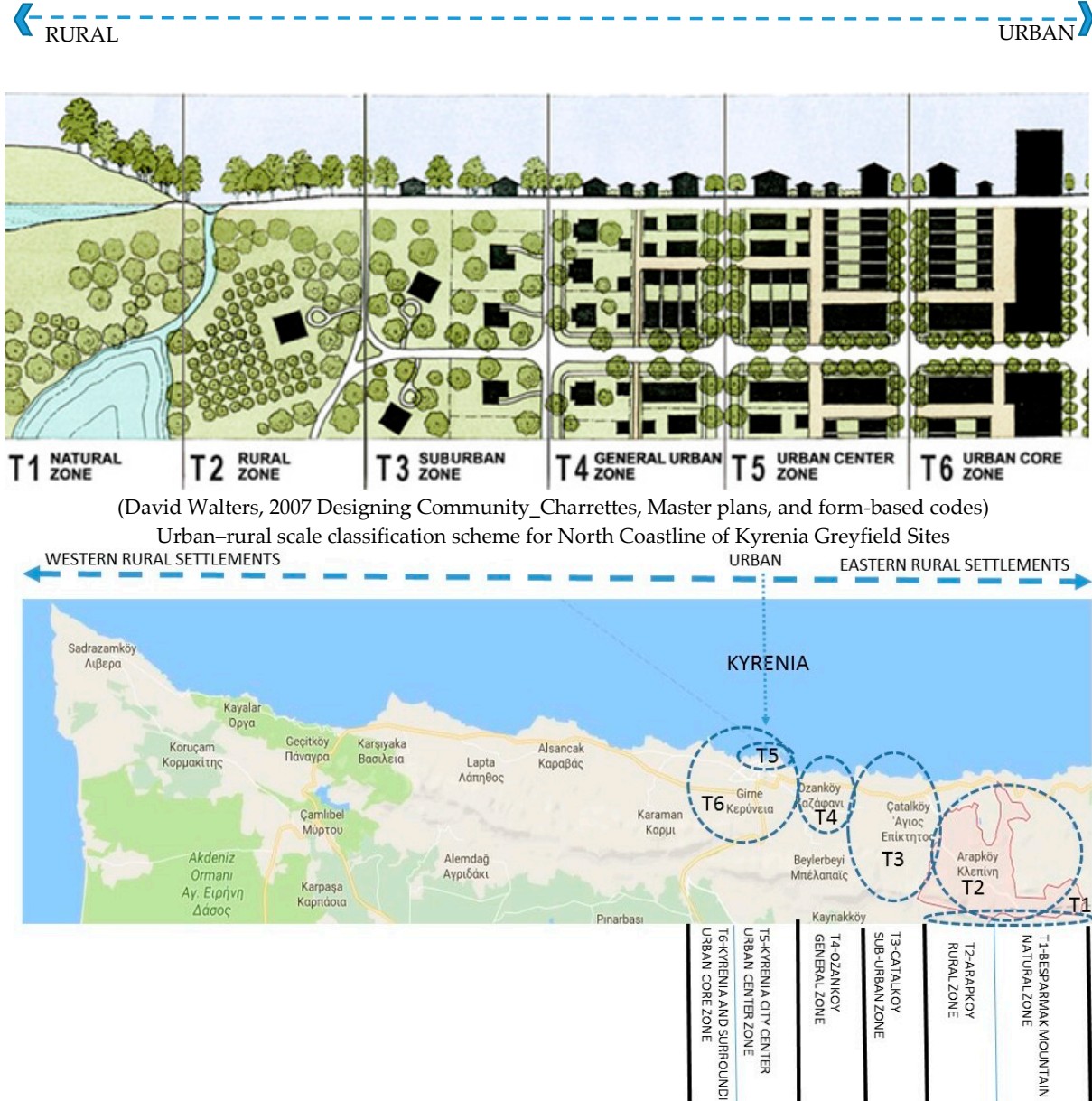

(David Walters, 2007 Designing Community_Charrettes, Master plans, and form-based codes)
Urban–rural scale classification scheme for North Coastline of Kyrenia Greyfield Sites

**Figure 7.** Urban–rural scale classification scheme: Kyrenia North Coastline. (Adapted from David Walters, 2007 Designing Community Charrettes, Master plans and form-based codes).

A Greyfield transect of 27.9 km in length was defined from the Kyrenia urban area to the eastern rural area and formed by six sections. The first section (T6) was in the city center close to Kyrenia and the last section (T1) was in the natural zone of the Besparmak Mountain area. The model transect zone diagram was based on the example of Kyrenia and its eastern settlements (Figure 8).

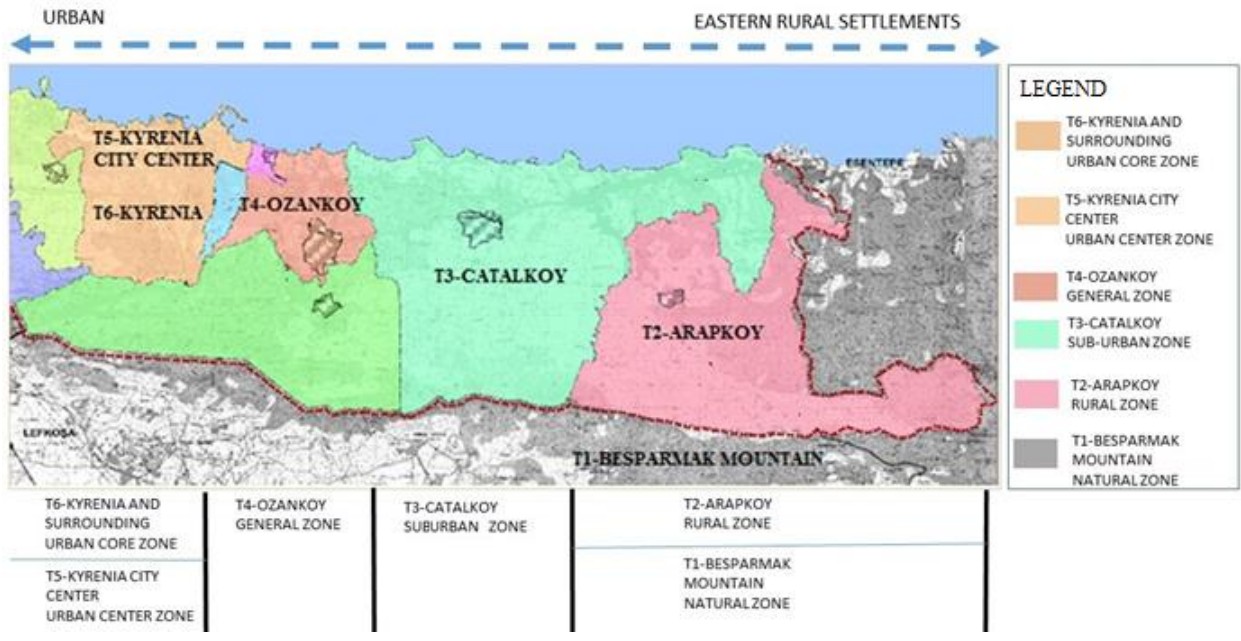

**Figure 8.** Urban–Rural Transect Theory. City of Kyrenia north coastal settlements.

The construction pattern, which was followed as a result of the urban–rural expansion, is presented. This enabled us to examine the state of the planning activities for the Greyfield sites (Table 5).

**Table 5.** Urban–rural scales classification of City of Kyrenia and north coastal settlements.

| Evaluation | Transect Theory Regions | | | | | |
|---|---|---|---|---|---|---|
| | T6-<br>Urban Core Zone<br>Kyrenia and<br>surrounding | T5-<br>Urban Centre Zone<br>Kyrenia City Centre | T4-<br>General Urban<br>Zone<br>Ozankoy | T3-<br>Suburban Zone<br>Catalkoy | T2-<br>Rural Zone<br>Arapkoy | T1-<br>Natural Zone<br>Besparmak<br>Mountain |
| | | Regions between T1 and T5 settlement zones are developing similarly as a result of urban expansion. | | | | |
| Density of Buildings | Dense building masses co-exist with historical and traditional textures. | Dense, multi-story buildings and single-story buildings co-exist. | Collective housing is observed intensely. | Collective housing is observed intensely. | There are constructions of mixed, traditional, and collective housing. | It is not open to settlement. |
| | | | They are developing and intertwined with each other. | | | |
| Status of Use | There are no Greyfield sites. | Although observed occasionally, Greyfield sites are not intense. | Due to its proximity to the city, Greyfield sites are not commonly observed. | It has Greyfield sites. | Greyfield sites are dense in new residential areas. | In some cases, a decision on housing construction can be made autonomously. |
| | | | | 71% of Greyfield sites are in rural areas, decreasing to 56% in the suburban zone. | | |
| Aim of Construction | Generally, the purpose of construction is oriented toward a built-to-sell architecture with the aim of profit-making. Collective housing areas gradually increase as the zone changes from the general urban zone, to the suburban zone to the rural zone, respectively. There is an increase in commercial areas as the zone changes to an urban core zone and an urban center zone. | | | | | |

As shown Table 5, the northern coastline of Kyrenia was divided into six zones based on the Transect Theory. The study is followed by a comparative analysis of the northern coastline of Kyrenia and east settlements based on the studies of [19,63].

*T1—Natural Zone (Five Finger Mountain Region)*: This zone includes areas that are not habitable for settlement due to its topography, hydrology, and vegetation but it covers areas

for wildlife and its habitat. However, in the north coastline settlements of Kyrenia, these zones are being opened for construction due to necessity, and their topography, hydrology, and biological diversity are being destroyed. These areas, which were determined as the borders of forest areas, are gradually becoming included in residential borders, thus causing natural areas to lose their ecological characteristics. Furthermore, since greenfield sites are being built in rural areas, at present, it is observed that these rural areas are beginning to lose their ecological characteristics.

*T2-Rural Zone (Arapkoy)*: This zone covers open fields, cultivated or sparsely settled areas. These are forests, agricultural lands, pastures, and irrigation areas. Several rural settlement areas in the north coastline of Kyrenia including fields, agricultural areas, and forests are opened for construction. They have become a part of the built-to-sell architecture. As is presented in Table 5, in natural zone areas where dense ecological characteristics can be observed, five-story architecture has been permitted. This creates the grounds for both regions to lose their features. There are high numbers of Greyfield sites in these areas; however, there are no legal regulations so far.

*T3-Sub Urban Zone (Catalkoy)*: Although this zone is similar to low-density settlement suburban areas, it differs from those because it hosts many occupational fields. Its vegetation is still natural. The blocks are very wide, and the roads are not compatible with the natural structure of the settlement. Suburban settlements in the north coastline of Kyrenia are residential areas that develop as a continuation of the high-density city and in particular have collective housing and apartments. Its vegetation is disappearing gradually, and there are settlements where infrastructure works that are incompatible with the natural structure are observed.

*T4-General Urban Zone (Ozankoy)*: This zone is denser compared to the zones mentioned above and primarily includes housing settlements. It has various residential architectural orders with mixed-use, including detached and terrace houses. The landscape is varied and medium-sized block islands define the streets. The traditional urban areas of Kyrenia were generally composed of detached or twin houses. However, today this has been combined with a high density of apartments and commercial buildings. The streets are similar throughout the city.

*T5-Urban Centre Zone (Kyrenia City Centre)*: Retail shops, offices, terrace houses, and apartments that are connected to the main roads constitute the center of this zone. It has narrow streets with wide pavements and trees. A high density of commercial buildings is observed in the Kyrenia city center, and apartments are the main living spaces; to a much lesser density there are also houses. It is not possible to describe the use of the mixed area with residential and commercial spaces; however, it is observed that the commercial sites are in zone density. Apart from the center of this zone, the pavements are narrow, and some streets do not have any pavements. Attempts to make the city center green were insufficient.

*T6-Urban Core Zone (Kyrenia and surrounding)*: This is the core of the city. The tallest buildings, and unique and various types of public buildings exist in this zone with very little or no natural vegetation. Detached houses and a high density of tall apartments with ground floors designed as commercial spaces co-exist in certain parts of this zone.

Roads are very crucial in the formation of the city center, its periphery, and suburban and rural areas. In the northern coastline of Kyrenia, the density of buildings, their use, facades, masses; junctions; public spaces; car parking lots; pavements; street silhouettes; lighting; characteristics, and the design of landscape elements and grey areas for each row evidence complication between them. The zones (T1–T6) were created based on the Urban–Rural Scaling Scheme, and starting from Kyrenia city, the characteristics of these zones have been observed to develop irregularly. When the Kyrenia city center is examined because construction permits are given based on the street width, the height of the buildings is irregular. Towards the suburban settlements, the effects of dense construction still can be observed. No regulation has been put in place yet to redevelop the Greyfield sites for public use. It is of utmost importance that the future planning studies consider urban, suburban,

and rural area planning as a whole, as in the Transect Theory. Moreover, planning should focus on the re-integration of ecological, economic, and social areas for public use, which is necessary for urban resilience.

Assessing statistical data for construction from 2004 to 2011, and when assessing the questionnaire findings, it can be argued that the problem of Greyfields is an unresolved problem that adversely has been affecting the region economically, ecologically, and socially. Being a touristic and coastal city, Kyrenia attracts many people as well as investors. There are privately owned universities, five-star hotels, and private schools located in this city, making it attractive for those who would like to have a built-to-sell business. Due to the unplanned developments and an increasing number of Greyfields, in 2016 a written decree was released regulating, and compared to before, limiting the conditions of building work. However, it did not provide a solution for the existing problem and caused adversity for those who would request to use their land for their personal use. The existing vacant houses and uncompleted construction problems have not been resolved. This is partly because these buildings are private investments; any intervention from the government in the form of regulation would cause a significant debate among stakeholders. The island of Cyprus was divided into two different societies and administrations due to political division. In this situation, two different currencies (Turkish lira and Euro) are used on the island, and the Turkish lira currency is used in the Kyrenia settlement. Another aspect is that due to international law and current political conflict in Cyprus, the property title deeds that are provided by the government are not recognized internationally. This has been reflecting the on-demand and sellability of these properties to the international market. Although there is a deadlock in terms of international recognition of the title deeds, there are still some interested international buyers, and that keeps the property prices in British pounds. In contrast, as of October 2010, the currency rate was more than 1 to 10. The Turkish lira is calculated over the current exchange rate. This situation makes properties out of reach for most of the local buyers, and no intervention from the government has lead to a sustained state of the existing problem.

Within the scope of the research, no data was found about the Subprime crisis (2008) of this situation.

## 5. Conclusions

Greyfield sites frequently emerge in suburban and rural areas. Within this context, Greyfield sites that exist in suburban and rural areas should be redeveloped for public use to ensure urban resilience. Transect Theory supports a planning strategy that starts from the urban areas towards rural; thus, it has a crucial role in guaranteeing the usability of Greyfield sites and urban resilience. Therefore, for future planning activities:

- Urban, suburban, and rural settlements are necessary to be considered as a whole.
- It is crucial to acknowledge that urban resilience in rural areas is vital in enabling urban areas as habitable.
- Greyfield sites are ecologically, economically, and socially problematic areas, and it is important to acknowledge the necessity of re-integration of Greyfield sites for public use.

Planning activities that would be carried out within this scope are very crucial in providing a method for bringing urban resiliency to the desired level. This would be enabled through intervention by regular tools that would create an opportunity to redevelop Greyfield sites, which emerge mainly in suburban and rural settlement areas.

Planning activities that would be carried out within this scope provide an opportunity to redevelop Greyfield. These planning activities are crucial in terms of providing a method that would bring urban resilience to the desired level from urban to rural, through regular intervention with tools. Within this context, it is expected that this study would contribute to the decision-making processes related to city planning, city design, and ecological planning.

As a result of the research, the Greyfield situation regarding the city of Kyrenia is as follows in Table 6.

**Table 6.** Kyrenia Greyfield Area ecological, economical, and social factors.

| | | |
|---|---|---|
| ECOLOGICAL | • Create pressure on natural environment. Prevent protection of environmental and ecological values<br>• Existence of housing above the area's carrying capacity leads to environmental pollution<br>• They create regional risk areas in term of urban resilience (i.e., natural disasters such as flood and earthquake, technological, and human-originated risks)<br>• Greyfield sites are obstacles to the development of urban and/or rural areas<br>• Greyfield sites limit the agricultural activities, which is a primary source of income<br>• The area is not managed with a plan | RESIDENT AND EXPERT(AGREE/STRONGLY AGREE) |
| | • Ensure protection and transfer of the region to future generations | RESIDENT AND EXPERT(DISAGREE/STRONGLY DISAGREE) |
| ECONOMICAL | • Greyfield sites are obstacles to the development of urban and/or rural areas<br>• Greyfield sites limit the agricultural activities, which is a primary source of income<br>• The area is not managed with a plan | RESIDENT AND EXPERT(AGREE/STRONGLY AGREE) |
| | • Greyfield sites contribute to the future of the regional economy | RESIDENT AND EXPERT(DISAGREE/STRONGLY DISAGREE) |
| | • Greyfield sites diversity the locals' income sources<br>• Contribute to the development of tourism | RESIDENT (AGREE)/EXPERT (STRONGLY DISAGREE) |
| SOCIAL | • Greyfield sites are obstacles to the development of urban and/or rural areas<br>• The area is not managed with a plan | RESIDENT AND EXPERT(AGREE/STRONGLY AGREE) |
| | • Ensure protection and transfer of the region to future generations | RESIDENT AND EXPERT(DISAGREE/STRONGLY DISAGREE) |
| | • Contribute to the development of tourism | RESIDENT (AGREE)/EXPERT (STRONGLY DISAGREE) |

Regarding the ecological issues, the residents and experts disagreed or strongly disagreed about "*ensuring protection and transfer of the region to future generations*". However, they agreed or strongly agreed on the following aspects: "*creating pressure on the natural environment*", "*preventing protection of environmental and ecological values*", "*existence of housing above the area's carrying capacity leads to environmental pollution*", "*they create regional risk areas in terms of urban resilience (i.e., natural disasters such as flood and earthquake, technological and human-originated risks)*", "*Greyfield sites are obstacles to the development of urban and/or rural areas*", "*Greyfield sites limit the agricultural activities which is a primary source of income*", and "*the area is not managed with a plan*".

Regarding the economic issues, the residents and experts disagreed or strongly disagreed that "*Greyfield sites contribute to the future of the regional economy*". However, they agreed or strongly agreed that "*Greyfield sites are obstacles to the development of urban and/or rural areas*", "*Greyfield sites limit the agricultural activities, which is a primary source of income*", and "*the area is not managed with a plan*'. In addition, "*Greyfield sites diversify the locals' income sources*" and "*contribute to the development of tourism*" was agreed upon by the residents, while the experts disagreed on these points.

Regarding social issues, the residents and experts disagreed or strongly disagreed that the Greyfields "*ensure protection and transfer of the region to future generations.*" However, they agreed or strongly agreed that "*Greyfield sites are obstacles to the development of urban and/or rural areas*" and "*the area is not managed with a plan*". In addition, the residents agreed that the Greyfields, "*contribute to the development of tourism*", while the experts disagreed.

As stated above by the residents and experts, in the evaluation of the ecological, economic, and social situation, it was revealed that Greyfield areas are problematic areas for urban resilience in the region. This study revealed the necessity of considering and evaluating the region as a whole, together with the Transect Theory, to ensure the durability of the urban–rural areas for future studies in the region. This study presents the effect of Greyfield areas on the environmental, social, and economic values within the framework of Transect Theory. Thus, a roadmap for reopening gray areas for public use has been proposed for use in future planning studies, which is a requirement for ensuring urban resilience. It is thought that this study will benefit city planners, architects, engineers, municipalities, government organizations, and researchers who will work in this direction.

**Author Contributions:** Conceptualization, V.A. and A.K.; methodology, V.A. and A.K.; software, V.A.; validation, V.A. and A.K.; formal analysis, V.A.; investigation, V.A.; resources, V.A.; data curation, V.A.; writing—original draft preparation, V.A.; visualization, V.A.; writing—review and editing, A.K.; supervision, A.K. All authors have read and agreed to the published version of the manuscript.

**Funding:** This research received no external funding.

**Institutional Review Board Statement:** Not applicable.

**Informed Consent Statement:** Not applicable.

**Data Availability Statement:** All data are available publicly as explained in the full article.

**Conflicts of Interest:** The authors declare no conflict of interest.

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
