# Peer review of "The Assessment of Greyfields in Relation to Urban Resilience within the Context of Transect Theory: Exemplar of Kyrenia–Arapkoy"

_sustainability, doi:10.3390/su15021181_

Round 1

Reviewer 1 Report

·        First five lines of the introduction have a different font size

·        The questionnaire part needs to be explained how the sample size was chosen based on the total number of residents.

·        The questionnaire survey presentation is very shallow and needs more in-depth to extract solid results.

·        Table 4 is not clear enough about how it was extracted from the study

·        Table 5 is not presentable and readable at all

·        The study was supposed to present the effect of greyfield sites on environmental, social, and economic values as mentioned in the abstract, but later on, it was not clear how it affects these three pillars.

·        Results and discussion part needs a lot of improvement in both contents and presentation to reflect what is mentioned in the abstract

Author Response

Amendment

REVIEWER 1 COMMENTS

REVIEWER 1

AUTHOR RESPONSE

Is the content succinctly described and contextualized with respect to previous and present theoretical background and empirical research (if applicable) on the topic?

ü   

Are all the cited references relevant to the research?

ü   

Are the research design, questions, hypotheses and methods clearly stated?

ü   

Are the arguments and discussion of findings coherent, balanced and compelling?

ü   

For empirical research, are the results clearly presented?

ü   

Is the article adequately referenced?

ü   

Are the conclusions thoroughly supported by the results presented in the article or referenced in secondary literature?

ü   

Reviewer 1

Author response

1

·   First five lines of the introduction have a different font size

ü   

we have been done

2

·   The questionnaire part needs to be explained how the sample size was chosen based on the total number of residents.

ü   

We would like to thank to the reviewer for this suggestion. The text methodology part is revised based on the comment.

3

·   The questionnaire survey presentation is very shallow and needs more in-depth to extract solid results.

ü   

we have been developed on recommendation

4

·   Table 4 is not clear enough about how it was extracted from the study

ü   

The text is revised based on the comment. We have been developed.

5

 · Table 5 is not presentable and readable at all

ü   

Tables have been updated.

6

 · The study was supposed to present the effect of greyfield sites on environmental, social, and economic values as mentioned in the abstract, but later on, it was not clear how it affects these three pillars.

ü   

The text is revised based on the comment. We have been made new table in text.

7

· Results and discussion part needs a lot of improvement in both contents and presentation to reflect what is mentioned in the abstract

ü                                                                         

The text is revised based on the comment.

Reviewer 2 Report

The authors developed research on an essential regional planning subject with a very interesting point of view, with the added advantage of the particular political and geographical context of Cyprus. It is not clear to the reviewer why the data used is essentially 6 years old, and the authors should be clearer on that justification, mainly because they refer to steps taken after 2016, which did not solve the issues with greyfields. In other words, it is a time-sensitive issue (data did not crystalize in 2016). Nevertheless, if some adjustments are made to the paper, its publication should be considered, due to its merit and originality.

Further comments/suggestions:

Line 13: use “(…) to analyze the greyfield problem (…)” instead.

Lines 14 and 84: use “(…) of Kyrenia, Cyprus, in relation (…)” instead.

Line 35: use “(…) it is also a (…)” instead.

Line 44: use “(…) suburbanization is crucial (…)” instead.

Line 65: use “(…) buildings that failed to succeed (…)” instead.

Line 68: use “(…) in Ramos (2011), greyfields are broadly defined (…)” instead.

Lines 72-73: use “(…) this study the ‘greyfiels’ term (…)” instead.

Line 85: include a paragraph briefly describing the structure of the paper at the end of section 1.

Line 87: the first paragraphs of section 2 could be included in section a numbered subsection (2.1 General methodology, for instance, and section 2.1 would be 2.2).

Line 92: use “(…) in reflecting the stakeholders’ (…)” instead.

Line 104: use “(…) for urban resilience were evaluated (…)” instead.

Line 125-126: use “(…) in 2007 was discussed (…)” instead.

Line 129: use “(…) about property issues (…)” instead.

Lines 150-151: use “(…) and cause problems (…)” instead.

Lines 156-157: use “In the CTBCA report, Cyprus Northern coastline (…)” instead.

Line 161: add the closing parenthesis.

Line 171-172:  use “(…) buildings, of those, 72 are commercial buildings.” instead.

Line 180: use “(…) not only cause irregular (…)” instead.

Line 199: use “(…) typologies, which are (…)” instead.

Line 202: how does this section appear here with this numbering?

Line 207: use “(…) is located, are as follows (…)” instead.

Lines 214-215: what do you mean by the sentence “Kyrenia East region has been completed in line with statistical data.”?

Lines 223-224: improve this figure.

Line 228: use “(…) rate, which is 26,9%, reaches the (…)” instead.

Lines 228-236: mention Figure 4 in this paragraph.

Line 237: revise “Situation” spelling.

Line 251: use “The construction information (…)” instead.

Lines 267-268: revise the sentence considering what you mean by rural. Rural environment? Rural dwellers?

Line 288: Figure 5 must be revised. The words rural and urban should be at the opposite ends of the line/arrows, for instance.

Line 289: use “Adapted from (…)” instead.

Line 296-297: use “(…) the percentage of area of each (…)” instead.

Line 300-301: use “Unplanned construction and distorted urban settlement in Kyrenia led greyfields to (…)” instead.

Line 306: use “(…) the region are referred as ‘experts’” instead.

Line 315: replace “mains water” with “water mains”.

Line 321: use “(…) were as follows: (…)” instead.

Lines 336-343: simply highlight the main aspects of Table 3. Do not describe the whole table.

Lines 349-377: just highlight the main aspects, such as overall statistics and main disagreements between experts and residents. Do not describe the whole table, as it is very clear with the colours the authors provided.

Line 384: use “(…) and cause obstacles to spatial development.” instead.

Line 397: the layout of table 5 must be improved. It is very hard to read. Once again, the words next to the arrows should be correctly placed.

Lines 477-480: please clarify: (i) whether the whole island of Cyprus is integrated into the Eurozone; (ii) what is the currency used in Kyrenia; and (iii) how did you calculate the mentioned currency rate (between which currencies, also).

Line 482: at the end of section 4.2, the authors could reflect on the relevance of the subprime crisis (2008) on the issue at hand. If data show that there is no influence, just state that.

Line 500: use “(…) which emerge mainly (…)” instead.

Author Response

Amendment

REVIEWER 2 COMMENTS

REVIEWER 2

AUTHOR RESPONSE

Is the content succinctly described and contextualized with respect to previous and present theoretical background and empirical research (if applicable) on the topic?

ü   

Are all the cited references relevant to the research?

ü   

Are the research design, questions, hypotheses and methods clearly stated?

ü   

Are the arguments and discussion of findings coherent, balanced and compelling?

ü   

For empirical research, are the results clearly presented?

ü   

Is the article adequately referenced?

ü   

Are the conclusions thoroughly supported by the results presented in the article or referenced in secondary literature?

ü   

Reviewer 2

Author response

1

The authors developed research on an essential regional planning subject with a very interesting point of view, with the added advantage of the particular political and geographical context of Cyprus. It is not clear to the reviewer why the data used is essentially 6 years old, and the authors should be clearer on that justification, mainly because they refer to steps taken after 2016, which did not solve the issues with greyfields. In other words, it is a time-sensitive issue (data did not crystalize in 2016). Nevertheless, if some adjustments are made to the paper, its publication should be considered, due to its merit and originality.

We would like to thank to the reviewer for this suggestion. The text is revised based on the comment. We have been researched new literature.

2

Line 13: use “(…) to analyze the greyfield problem (…)” instead.

ü   

We have been done

3

Lines 14 and 84: use “(…) of Kyrenia, Cyprus, in relation (…)” instead.

ü   

We have been done

4

Line 35: use “(…) it is also a (…)” instead.

ü   

We have been done

5

Line 44: use “(…) suburbanization is crucial (…)” instead.

ü   

We have been done

6

Line 65: use “(…) buildings that failed to succeed (…)” instead.

ü   

We have been done

7

Line 68: use “(…) in Ramos (2011), greyfields are broadly defined (…)” instead.

ü   

We have been done

8

Lines 72-73: use “(…) this study the ‘greyfiels’ term (…)” instead.

ü   

We have been done

9

Line 85: include a paragraph briefly describing the structure of the paper at the end of section 1.

ü   

The text is revised based on the comment.

10

Line 87: the first paragraphs of section 2 could be included in section a numbered subsection (2.1 General methodology, for instance, and section 2.1 would be 2.2).

ü   

The text is revised based on the comment.

11

Line 92: use “(…) in reflecting the stakeholders’ (…)” instead.

ü   

We have been done

12

Line 104: use “(…) for urban resilience were evaluated (…)” instead.

ü   

We have been done

13

Line 125-126: use “(…) in 2007 was discussed (…)” instead.

ü   

We have been done

14

Line 129: use “(…) about property issues (…)” instead.

ü   

We have been done

15

Lines 150-151: use “(…) and cause problems (…)” instead.

ü   

We have been done

16

Lines 156-157: use “In the CTBCA report, Cyprus Northern coastline (…)” instead.

ü   

We have been done

17

Line 161: add the closing parenthesis.

ü   

We have been done

18

Line 171-172:  use “(…) buildings, of those, 72 are commercial buildings.” instead.

ü   

We have been done

19

Line 180: use “(…) not only cause irregular (…)” instead.

ü   

We have been done

20

Line 199: use “(…) typologies, which are (…)” instead.

ü   

We have been done

21

Line 202: how does this section appear here with this numbering?

ü   

The text is revised based on the comment.

22

Line 207: use “(…) is located, are as follows (…)” instead.

ü   

We have been done

23

Lines 214-215: what do you mean by the sentence “Kyrenia East region has been completed in line with statistical data.”?

ü   

The text is revised based on the comment.

24

Lines 223-224: improve this figure.

ü   

We have been done

25

Line 228: use “(…) rate, which is 26,9%, reaches the (…)” instead.

ü   

We have been done

26

Lines 228-236: mention Figure 4 in this paragraph.

ü   

We have been done

27

Line 237: revise “Situation” spelling.

ü   

We have been done

28

Line 251: use “The construction information (…)” instead.

ü   

We have been done

29

Lines 267-268: revise the sentence considering what you mean by rural. Rural environment? Rural dwellers?

ü   

We have been done

30

Line 288: Figure 5 must be revised. The words rural and urban should be at the opposite ends of the line/arrows, for instance.

ü   

The text is revised based on the comment.

31

Line 289: use “Adapted from (…)” instead.

ü   

We have been done

32

Line 296-297: use “(…) the percentage of area of each (…)” instead.

ü   

We have been done

33

Line 300-301: use “Unplanned construction and distorted urban settlement in Kyrenia led greyfields to (…)” instead.

ü   

We have been done

34

Line 306: use “(…) the region are referred as ‘experts’” instead.

ü   

We have been done

35

36

Line 315: replace “mains water” with “water mains”.

ü   

We have been done

37

Line 321: use “(…) were as follows: (…)” instead.

ü   

We have been done

38

Lines 336-343: simply highlight the main aspects of Table 3. Do not describe the whole table.

ü   

The text is revised based on the comment.

39

Lines 349-377: just highlight the main aspects, such as overall statistics and main disagreements between experts and residents. Do not describe the whole table, as it is very clear with the colours the authors provided.

ü   

The text is revised based on the comment.

40

Line 384: use “(…) and cause obstacles to spatial development.” instead.

ü   

We have been done

41

Line 397: the layout of table 5 must be improved. It is very hard to read. Once again, the words next to the arrows should be correctly placed.

ü   

The text is revised based on the comment.

42

Lines 477-480: please clarify: (i) whether the whole island of Cyprus is integrated into the Eurozone; (ii) what is the currency used in Kyrenia; and (iii) how did you calculate the mentioned currency rate (between which currencies, also).

ü   

The text is revised based on the comment.

43

Line 482: at the end of section 4.2, the authors could reflect on the relevance of the subprime crisis (2008) on the issue at hand. If data show that there is no influence, just state that.

ü   

The text is revised based on the comment.

44

Line 500: use “(…) which emerge mainly (…)” instead.

ü   

We have been done

Reviewer 3 Report

The authors have submitted an article based on the study of "The Assessment of Greyfields in Relation to Urban Resilience 2 within the Context of Transect Theory: Exemplar of Kyrenia- 3 Arapkoy"

Despite the fact, the study is limited to a very specific area. The analysis provided deserves attention. Overall, the article is well-written and supported by graphs and figures.

However, the following points must be amended:

- figure 2: where the graph is taken from? The citation is missing.

- the colour code used in figure 4 is not clearly explained. A legend is necessary. 

Author Response

Amendment

REVIEWER 3 COMMENTS

REVIEWER 3

AUTHOR RESPONSE

Is the content succinctly described and contextualized with respect to previous and present theoretical background and empirical research (if applicable) on the topic?

ü   

Are all the cited references relevant to the research?

ü   

Are the research design, questions, hypotheses and methods clearly stated?

ü   

Are the arguments and discussion of findings coherent, balanced and compelling?

ü   

For empirical research, are the results clearly presented?

ü   

Is the article adequately referenced?

ü   

Are the conclusions thoroughly supported by the results presented in the article or referenced in secondary literature?

ü   

Reviewer 3

Author response

1

The authors have submitted an article based on the study of "The Assessment of Greyfields in Relation to Urban Resilience 2 within the Context of Transect Theory: Exemplar of Kyrenia- 3 Arapkoy"

Despite the fact, the study is limited to a very specific area. The analysis provided deserves attention. Overall, the article is well-written and supported by graphs and figures.

We would like to thank to the reviewer for this suggestion.

2

- figure 2: where the graph is taken from? The citation is missing.

ü   

It has been added.

3

- the colour code used in figure 4 is not clearly explained. A legend is necessary. 

ü   

It has been added.

4

5

6

7

Reviewer 4 Report

The paper discusses the topic of greyfields assessment in relation to urban resilience and aims to present a road map for the redevelopment ofgreyfields for public use to be used in future planning activities. 

In my opinion, the paper lacks clear structuring of content and a deeper framing with respect to the topic addressed. 

Specifically, I think that a clear explication of the research methodology is needed, providing information about the research design. 

In addition, I believe that the paper lacks theoretical background related to the theme of resilience and greyfields, and to the theme of "Transect Theory." Both are not outlined and explained in detail by referring to the existing literature. With respect to the topic of greyfields, Australian and American literature is partially cited, from which, however, arises the question: why consider and cite only this literature if the case study is a European case?

Finally with respect to the results I think it is necessary to better highlight the road map for the redevelopment of greyfields for public use.

I believe that the work needs to be extensively revised.

Author Response

Amendment

REVIEWER 4 COMMENTS

REVIEWER 4

AUTHOR RESPONSE

Is the content succinctly described and contextualized with respect to previous and present theoretical background and empirical research (if applicable) on the topic?

ü   

Are all the cited references relevant to the research?

ü   

Are the research design, questions, hypotheses and methods clearly stated?

ü   

Are the arguments and discussion of findings coherent, balanced and compelling?

ü   

For empirical research, are the results clearly presented?

ü   

Is the article adequately referenced?

ü   

Are the conclusions thoroughly supported by the results presented in the article or referenced in secondary literature?

ü   

Reviewer 4

Author response

The paper discusses the topic of greyfields assessment in relation to urban resilience and aims to present a road map for the redevelopment of greyfields for public use to be used in future planning activities. 

1

In my opinion, the paper lacks clear structuring of content and a deeper framing with respect to the topic addressed. 

ü   

We would like to thank to the reviewer for this suggestion. The text is revised based on the comment. We have been made clear structuring of content and a deeper framing with respect to the topic addressed.

2

Specifically, I think that a clear explication of the research methodology is needed, providing information about the research design. 

ü   

We have been developed research methodology.

3

In addition, I believe that the paper lacks theoretical background related to the theme of resilience and greyfields, and to the theme of "Transect Theory." Both are not outlined and explained in detail by referring to the existing literature. With respect to the topic of greyfields, Australian and American literature is partially cited, from which, however, arises the question: why consider and cite only this literature if the case study is a European case?

ü   

 The text is revised based on the comment. We have been found new researches about this suggestion. And we have been developed literature review.

4

Finally with respect to the results I think it is necessary to better highlight the road map for the redevelopment of greyfields for public use.

ü   

The conclusion has been revised based on the comment.

Round 2

Reviewer 1 Report

all the earlier comments were corrected and the paper developed a lot

Author Response

Thank you for your interest in the article editing process. In light of the information we have obtained from you, we have learned a lot. Thanks again...

Reviewer 2 Report

The authors made a good effort to improve their manuscript. This reviewer’s comments were answered.

However, considering the voluntary effort reviewers make to revise manuscripts, the authors could have the kindness to indicate where the changes were placed in the revised manuscript, or provide more than “done” when a question is made (this reviewer is not referring to simple English corrections).

This reviewer would like to congratulate the authors on their paper.

Author Response

(The authors gave the same response as above.)

Reviewer 4 Report

The authors have partially absolved previous reviews.

It seems unusual to find a theoretical description of "Transect Theory" in the results section (4.3), when the purpose of the paper is not a literature review. I suggest to the authors to consider the option of including this description in section 2 (Materials and Methods) and/or implementing the methodology section (2.1) in these terms, allowing the reader to understand how this analytical method is used to perform the analysis. I believe that this is an element still missing in the paper and it constitutes an important component for understanding the research. Eventually, it might be helpful to support this explanation with a diagram or graphical representation.

I highlight a formatting error in the text at lines 340 - 344.

I suppose that line 492 is a repetition of the caption in Figure 7.

I also believe that the punctuation in Section 4 needs to be checked (Ex. "Therefore, for future planning activities;", "Regarding Ecological issues; residents and experts do disagree [...]")

Author Response

Thank you for your interest in the article editing process. In light of the information we have obtained from you, we have learned a lot. Thanks again...

Reviewer 4

Author response

1

It seems unusual to find a theoretical description of "Transect Theory" in the results section (4.3), when the purpose of the paper is not a literature review. I suggest to the authors to consider the option of including this description in section 2 (Materials and Methods) and/or implementing the methodology section (2.1) in these terms, allowing the reader to understand how this analytical method is used to perform the analysis. I believe that this is an element still missing in the paper and it constitutes an important component for understanding the research. Eventually, it might be helpful to support this explanation with a diagram or graphical representation.

ü   

We would like to thank the reviewer for this suggestion. The text methodology part is revised based on the comment.

2

I highlight a formatting error in the text at lines 340 - 344.

ü                                                                         

The text is revised based on the comment.

3

I suppose that line 492 is a repetition of the caption in Figure 7.

ü                                                                         

The text is revised based on the comment.

4

I also believe that the punctuation in Section 4 needs to be checked (Ex. "Therefore, for future planning activities;", "Regarding Ecological issues; residents and experts do disagree [...]")

ü                                                                         

The text is revised based on the comment.

5
